# The Multiplicative Effect Interaction between Outdoor Education Activities Based on the Sensory System

Veronica Mindrescu [1], Gabriel Simion [1,*], Ioan Turcu [1,*], Cristian Catuna [2], Dan Gheorghe Paun [2] and Florentina Nechita [1,*]

1 Faculty of Physical Education and Mountain Sports, Transylvania University Brasov, 500036 Brasov, Romania
2 Faculty of Physical Education and Sports, Spiru Haret University, 030045 Bucharest, Romania
* Correspondence: gabriel.simion@unitbv.ro (G.S.); ioan.turcu@unitbv.ro (I.T.); florentina.nechita@unitbv.ro (F.N.); Tel.: +40-722-416-423 (G.S. & I.T.)

**Abstract:** *Background:* The present paper initiates the introduction of physical education activities within the Transylvania University of Brașov, aiming at a new strategy. The purpose is identifying the level of knowledge and the level of perception regarding the extent to which outdoor activities are viewed and implemented, and the effects that the latter has on them by tracking certain variables of a sensory nature, comprised of visual, auditory, kinesthetic and digital areas. *Methods:* This research aims to analyze if the type of sensory channel is influenced by the type of sport and the environment (urban/rural) of that practiced sport. We also analyzed the benefits offered by different sports and if these benefits influence the type of activity. Data were collected using an online survey, a questionnaire, using a Likert scale, with subjects having to choose between multiple answers. In addition, data were allocated and reviewed based on a sample of 100 students who have the habit of practicing outdoor activities. *Results:* The results of the study significantly show that the objectives were met and as such it can be concluded that outdoor activities, from the point of view of perception systems, can be classified according to the method of ordering the rank of activities by the dominant kinesthetic sensory channel.

**Keywords:** multiplicative effects; interaction; outdoor education; sensory systems





## 1. Introduction

Outdoor education as a form of education comprises, alongside the learning factor, its experiential aspect and its setting, because it is primarily conducted outside, as well as the concept of senses and how they are perceived and used in all their characteristics and domains [1]. Education effectuated outside, in a natural environment, has started to put quite a lot of emphasis on the perception of the senses, whereas this perception is made based on different feelings experienced outside or based on factual observation and the dualistic aspect of the cause-effect relationships arisen within the undertaking of different outdoor activities [2].

Referring to outdoor education activities, it is necessary to point out that the major objective, almost unique, is acknowledging and surpassing one's limits; victory, as an objective that comes from the accomplishment of each individual, is also pointed out [3]. This need represents an energizing part of the outdoor education activity, leading to obtaining a certain high level of self-esteem. The benefits of this accomplishment are first of all the ideal and then the material. If the programs of outdoor education have an organized, specific structure that the educators can provide, then the outdoor learning notion becomes not only a point of challenge for the students when undertaking different activities but also builds changes in oneself, leading to growth, personal development, and self-confidence [4]. The outdoor activities and sports are mostly practiced gaining a high level of fitness and good health. Another important motivation is having fun near

family, friends, and colleagues [1]. They love the experience of feeling excitement and adventure due to the adrenaline release. People that practice outdoor activities have the benefit of observing the scenic beauty and being close to nature. Talking about outdoor sports, the most important gain is developing specific skills, abilities and gaining a sense of self-confidence [2–4].

The need to wisely use open, free spaces is a product of the culture and education, life evolution, and contemporaneous civilizations. Today, when work becomes less and less physical and more and more intellectual, free time spent in different ways and forms, as a necessary complementarity to the routine and stressful activities from closed spaces, becomes the most valuable time asset outside productive work [5].

In the course of the contemporary development of life, within the last couple of years with the unfolding of the COVID-19 pandemic and the lockdowns, parents have started to work more from home, thus leading to children staying at home and performing activities inside. This sedentariness has meant fewer and fewer trips outside, losing the connection with nature for a while, becoming more engaged in things primarily effectuated in confined spaces, and experiencing a high level of stress for both parents and children [6–8].

The delusion of modern technological methods, such as the television or the computer, has led to a sedentary life among children and adults. Lately, a worrying increase in the number of overweight or anemic people has been registered both among adults and children who are attracted to video games and movies and forget, with their parents' consent, about outdoor games and walks in nature [9–11].

Though these are useful for the individual's intellectual development, they do not always offer everything necessary to develop a balanced and harmonious life. The specialized knowledge and situations where one refers to multi-disciplinary cooperation all can prove that a major part is played by outdoor activities, and one presumes that their effectuation makes it possible to highlight a superior age branch of the student's main sensory system. The need to develop the human personality through outdoor activities is believed to represent the broad field of human experiences dedicated to forming healthy characters; in this way, they can build, keep and transmit further along with a much cleaner, more beautiful, and healthier world [12–15].

Outdoor education activities are interactive activities that simulate real situations and involve resolving certain tasks. "An experiential way of learning that involves using all senses" takes place generally, but not exclusively, through exposure to the natural environment through outdoor education activities. They involve all three fields: physical, psychological, and emotional. Consequently, the participants assimilate a series of abilities and skills that contribute to improving personal performances; furthermore, when team members become aware of the obstacles that harden the teamwork, all of this contributes to improving the team's performance both within the exercise and in "real life" [16].

The main objective of outdoor activities is to encourage the development of certain personality traits with broad social acceptance: initiative, perseverance, optimism, willingness, organizing skills, courage, and special organizing skills. These characteristics arise from spending time performing outdoor activities and are constantly developing when they are constantly effectuated. Playing outside or engaging in different outdoor activities creates relationships, develops the social connections between peers, and develops motor skills, while also generating more learning possibilities in the natural environment and encapsulating a healthier mindset in all the upcoming challenges society might bring [17–20].

Moreover, in analyzing the personality traits, one must bear in mind the dynamics that characterize them and the sudden or gradual transformations that have taken place at their level. The dynamic character of free time activities also determines differentiated attitudes towards the effectuated activity and differentiated relationships between the group members. In other words, an individual does not manifest the same behavior or attitude in connection to the effectuated activity of each member of that respective group [21].

## 2. Review Research of Web of Science—Articles 2018–2020

The research design was defined by the following topics: "outdoor education and sports" (Topic) and 2018–2020 (Year Published) and Articles or Review Articles (Document Types) and all categories from (Web of Science Categories). As a result, only 360 papers were returned. We selected only 40 papers. The rest of the 320 was in another language than English, or do not offer the full text, or analyses collateral to the outdoors sports theme. Figure 1 shows the authors' high visibility of most respective papers and the link between themes, and publication year.

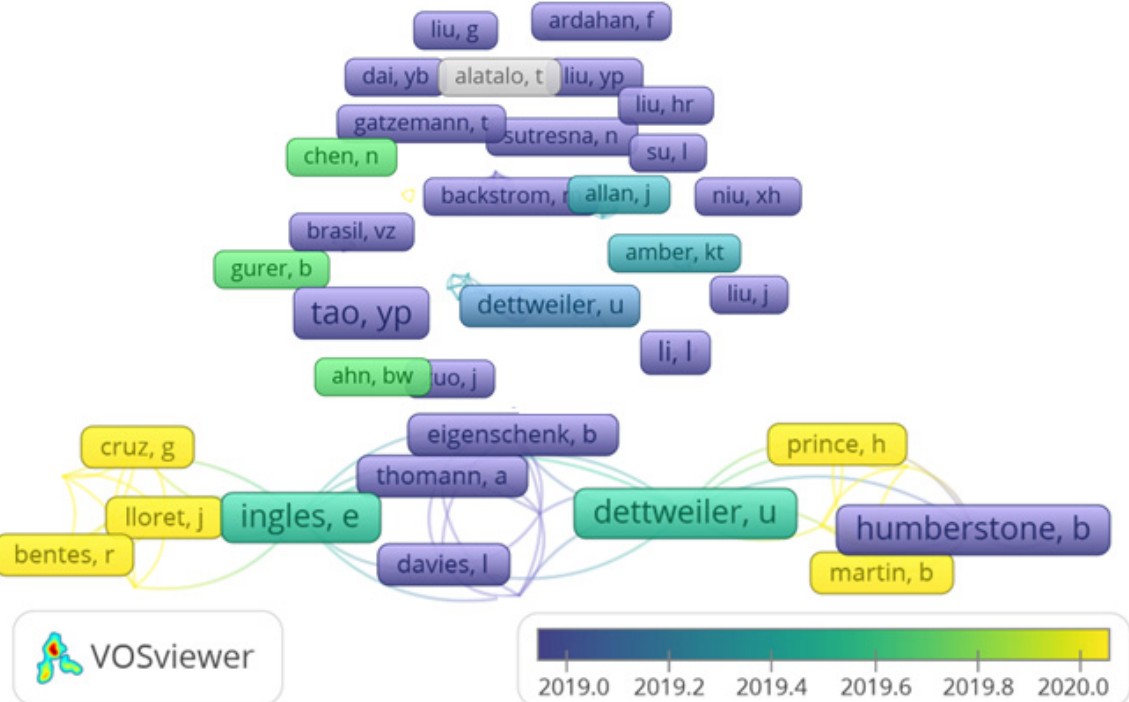

**Figure 1.** Clusters of papers published from 2018 to 2020 (Source: VOSviwer version 1.6.18 (May 2022).

We decided to use VOSviewer Software for a professional panorama regarding scientific papers published in the last years grouped in three main clusters. This image helps us to understand the topics discussed by the main authors, and the interrelation between them.

The social benefits of practicing sport and teaching within the natural environment are detailed by Eigenschenk, in a very large study with students from six European countries. They assumed that the main benefits of outdoor education are physical health, mental equilibrium, education, citizenship abilities, attitude, crime mitigation, and anti-social behavior [22,23]. Another author conducted a similar study in Viana do Castelo (Portugal) for school-age children and adolescents that were the beneficiary of nautical activities. Besides the advantages presented above, they also discuss environmental awareness [24].

Within the non-formal educational programs, "outdoor education" represents a learning process that takes place outside. Outdoor education includes environmental education to enhance the sense of responsibility towards the natural frame of outdoor activities; adventure education by setting the scene properly from the social and cultural point of view as to engulf the needs and desires of children as well as the newly established relationships within the context of outdoor activities; the benefits of camping as a useful tool for developing new relationships and for practicing perceptive feeling during different situations; outdoor therapy activities and certain aspects of outdoor recreation; notions that have grown so much in the last decades; notions that trigger a healthier state of mind and well-being. Thus, one considers that practicing outdoor education activities can be-

come a method of teaching youngsters, these activities being elaborated with a training purpose; for example, the group cooperation activities cultivate honesty, respect for the other members' feelings and rights, care for others, and self-discipline [25–28].

Recent research has shown the importance of outdoor education on character development, collaborative social behavior, and health for all the participants in nature activities. Concurrently, the subjects get to know nature, its laws, and balance based on elements and a whole. The subjects' nature experience stimulates creativity, engaging the body and the mind and amplifying a better memory of their previous experiences and activities; these qualities are generally inhibited by the closed, artificial, and stressful environment of cities [29].

As scientific research, outdoor education drifts its education methods away from quantitative research dominated by the descriptive and the statistical characteristic to research based on hermeneutics, that is on interpreting and analyzing the experimental facts and data, thus accomplishing the link between the objective and the subject of the research as a direct link [30].

The characteristics of the motile acts are kinesthetically different, depending on the effectuated physical activity: some are extremely complex involving a thorough control of the organism and its segment, while others separately follow certain qualities, skills, or motile abilities. The optimal level of motivation for these complex motile acts requires extremely sensitive coordination and the intervention of intellectual processes. To solve complex tasks, one needs moderate motivation, while for simple tasks one needs high motivation. The ones engaging in outdoor education activities must know very well their own psychological and motile characteristics for their respective age [31].

Man is quite reasonably structured: to know the surroundings he has at his disposal a universal set of sensory organs. Everybody can hear, see, feel and understand. It is true, that few people pay attention to the fact that when someone interacts with the outside world, in most cases, they do not use all the tools provided by nature, but only the selected ones. Such a holistic approach gives man the opportunity to feel and assimilate a different kind of learning experience, an event that enhances the senses, the personality, and the motor skills of the individual [32].

The research evaluates the visual, audio, kinesthetic, and digital characteristics as one of the most important in elaborating a person's psychological portrait. Knowing his/her psychological type will not only help to get along with his /herself but will also simplify the process of interaction with other people.

For some it is more important to receive the auditory information, while for others it is more convenient to see the information with their own eyes; nevertheless, others cannot learn anything unless it is projected in their personal life, while for others a common language can be found if one speaks in the language of strict logic. This is to say that everybody has a certain specific way of perceiving information and depending on that, everybody shares visual, auditory, kinesthetic, and digital information [32].

To identify to what extent the proportion between outdoor education activities and the sensory system exists, as well as the interaction between these two elements, these relations are expressed through practical and preferential methods such as native forms as the need for movement or through preponderantly social and educational methods such as the need for self-knowledge, sense of risk and its calculation, as well as the habit of practicing movement [33].

The study also presumes that the educational formers are the ones who have the role of appreciating the harmonious social relationships of the youngsters as a result of their participation in outdoor education activities. In this context, experiential learning cannot be permanently systematic or uniform; it addresses the different characters and personalities of each participant. These skills are developed in time and with maximum attention on behalf of educators, for example, through group cooperation activities that cultivate honesty, respect for the other members' feelings and rights, care for others, and self-discipline, as ethical values. Accomplishing this educational model also requires specific strategies,

through which the students can learn to communicate with each member of the group, take responsibility, and accept the differences and the compromises within a group. The physical education teachers must know the fact that these subjects are sometimes below expectations in this area, not because they do not want to participate in these activities but because they do not know, with gradual teaching of these skills being the natural step. This happens also because some subjects are not aware of the collaborative values within a group at home or in other contexts and because these values have never been consolidated [33,34]. As a consequence, teachers must not expect the participants to learn these values at a fast and automatic pace. Learning these collaborative skills is done just like learning any other skills. If the subjects do not understand the group collaborative concepts and practice, educators must never direct them in solving their misunderstandings or disputes outside the group, but through outdoor activities that generate beneficial multiplicative effects in the individual, group, and social areas [30,35].

## 3. Materials and Methods

This research aims to analyze if the type of sensory channel is influenced by the type of sport and the environment (urban/rural) of that practiced sport. We also analyzed the benefits brought by different sports and if these benefits influence the type of activity (visual, auditory, kinesthesis, and digital). The data have been collected through the help of an online survey, a questionnaire, using the Likert scale with the subjects having to choose from multiple answers; therefore, a multiple-choice type of questionnaire. Furthermore, the data have been allocated and reviewed based on a sample of 100 students from the Transylvania University of Brasov, students that are in the habit of practicing outdoor activities.

The sample is representative. Data were analyzed with Microsoft Excel 365 in the first stage. In the second stage, we used SmartPLs Software version 3.0 for a factor analysis (CFA), which measures the impact of factors on the dependent explanatory construct.

### 3.1. Stage I—Perception Analysis Based on the Likert Scale
3.1.1. Stage I Hypothesis

**Hypothesis 1 (H1).** *The research hypothesis: one presumes that no matter the outdoor education program or the motives related to the sensory system for students, the degree of satisfaction in obtaining performance and surpassing their limits is important.*

3.1.2. Stage I Method

The study has as a purpose the identification of the student's preferred means of outdoor education activities through the dominant sensory system. Each student was stimulated based on his/her perceptions—visual, auditory, kinesthetic, and digital—to appreciate the effectuated outdoor education activities within a practical stage in a specialized center for these kinds of activities.

One of the methods used in the research was the Likert scale, having as an objective the appreciation of outdoor activities included in 4 modules that have been effectuated within the practical stage.

The sensory analysis method was conducted through a questionnaire that has as an appreciation system of the measurement of the student's satisfaction towards the practical stage effectuated means and methods. The scale had 5 degrees, which indicated the intensity of the agreement or disagreement of the participants to such outdoor education activities. The Likert scale had the 5 following degrees: Total agreement (+2), Agreement (+1), Indifference (0), Disagreement (−1), and Total disagreement (−2). [36]

Another method of research is the rank ordering method [37] to classify the proposed outdoor activities for the students to effectuate them based on the intensity of a single sensory characteristic that is the preferred one; the objectives were the following:

- Facilitating a more precise evaluation;
- Facilitating the appreciation through acceptance of the proposed outdoor activities.

The research used the same sample of 100 students that have been asked to express their preferences regarding the proposed outdoor activities; this is a fast and quite accurate method in evaluating certain complex characteristics such as perceiving the activities through the preferred sensory canal. The students were invited to experiment with the outdoor activities in a practical stage in a specialized center and they were asked to express their satisfaction and preferences related to this experience. The 100 students grouped in 4 clusters of 25 students have effectuated the following outdoor activities:

- Trekking and hiking;
- Mountaineering hiking;
- Orienteering and movement in different types of the field;
- Climbing a fake wall;
- Rappel;
- Zipline;
- Rope garden;
- Touristic orientation;
- Rafting, kayak canoeing;
- Mountain biking;
- Building a boat;
- Writing maps;
- Static activities based on communication.

All these activities have been proposed with the stimulation of the 4 sensory canals as a main aim:

- Activity 1 (centered on the dominant visual sensory canal)—activities that have presumed excursions and hiking in which the students were asked to contemplate nature, to effectuate activities with precise tasks such as finding a hidden treasure and orientating on the field to write a map; in these activities one presented the beauties of nature, knowing different species of plants and birds, things that required evaluating the information given, especially through the preferred sensory canal—the visual one.
- Activity 2 (centered on the dominant auditory sensory canal)—these activities presumed precise tasks without useless details; their solution also presumed communication guided mainly by the auditory sensations.
- Activity 3 (centered on the dominant kinesthetic sensory canal)—the activities have presumed to solve the tasks based on experimenting with certain physical sensations, perceiving pain, coldness, hotness, the difficulties of solving tasks in different ways, solving certain difficult motile tasks such as the rope garden, climbing rocks, rappelling and undertaking expeditions for surviving.
- Activity 4 (centered in the dominant digital sensory canal)—these activities required the management of logic and thought process with clear planning; they were stressful activities such as building a boat from different objects and experimenting this on water and building certain objects with materials from nature without affecting nature, paths or maps.

### 3.1.3. Stage I Results

The students' opinions regarding the 4 types of activity are presented in Table 1. The score regarding each type of activity was calculated with the formula

$$An = [NoA \times (+2) + NoA \times (+1) + NoA \times 0 + NoA \times (-1)]: 100 \qquad (1)$$

where An is the name of the activity and NoA represents the number of answers for each type of activity.

**Table 1.** The Likert Scale the students' appreciation.

| Activity | Total Agreement (+2) | Agreement (+1) | Indifference 0 | Disagreement (−1) | Total Disagreement (−2) |
|---|---|---|---|---|---|
| Activity 1—sensory | 72 | 26 | 1 | 1 | 0 |
| Activity 2—auditory | 70 | 27 | 1 | 2 | 0 |
| Activity 3—kinesthetic | 75 | 23 | 1 | 1 | 0 |
| Activity 4—digital | 62 | 29 | 5 | 4 | 0 |

The global score of this study is calculated with the formula $\frac{\sum_{i=1}^{n} An}{n}$, where n = 4 in our case, the 4 types of activities from Table 1.

On a scale from −2 to +2, the global score was 1.65 positive value, and the most appreciated module was module 3 with a score of 1.73 (Figure 2).

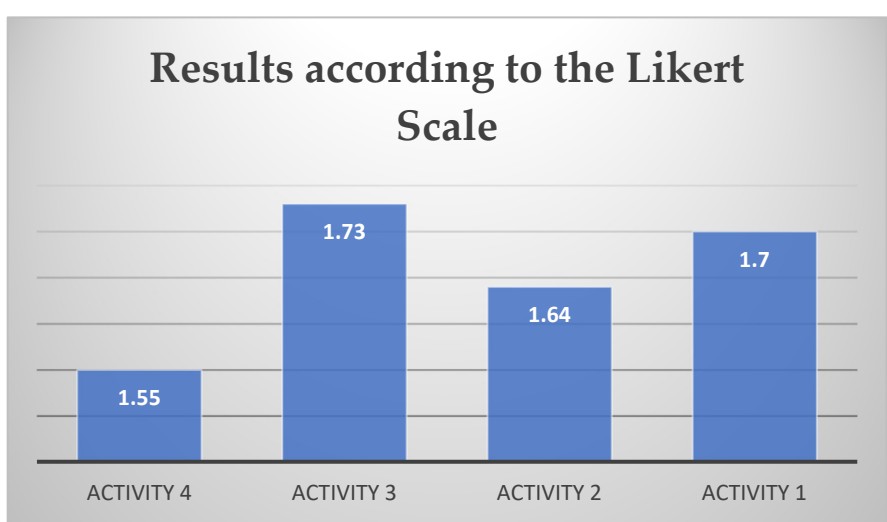

**Figure 2.** The research results according to the Likert Scale.

From the data research analysis, one can see that the overall image is positive, and the participant students have expressed through their options module 3 as a winner, the module centered on the dominant kinesthetic sensory canal with a score of 1.78, followed by module 2 centered on the dominant visual sensory canal with a score of 1.73.

Analyzing the research study results after the rank ordering method, one can conclude that the students preferred the activities from module 3 (centered on the kinesthetic sensory canal), with a first-place score of 1.73 followed by a tight score from module 1 activities (centered on the visual sensory canal) with 1.7. The third place was occupied by module 2 with auditory activity and a score of 1.64. The last place was occupied by module 4 with digital activity and a score of 1.55.

Thus, one can also confirm H1 by the results obtained: students' satisfaction and appreciation regarding outdoor activities are positive. The better the ability is to work with these canals the better one can communicate with oneself and those around them. A first step in training this ability is discovering the preferred sensory canal.

After closely observing the students' preferences, based on their behavior and observation charts, the ones who preferred module 3, with a score of 1.73 centered on the kinesthetic sensory canal, were aware of how they felt things and emotional reactions, and they preferred to gather information through touching, gesturing and smelling. They were willing to experiment with the emotion of the unknown and the need for movement, which contribute to improving and developing the collaborative coordination through activities that had as tasks moving on a varied field, without tracks, maintaining balance depending on the effectuated path, developing courage, the needs to affirm oneself, a

better understanding of others, and a better way of showing respect for the other's abilities and skills.

The students who marked as preference the activities from module 1 (degree score of 1.7 centered on the dominant visual sensory canal) stood out by solving tasks such as spatial orientation, using clear schemes, and often using the words "beautiful" or "ugly" in correlating the information provided by nature through images.

The ones opting for module 2, degree score of 1.64 (centered on the dominant auditory canal), resolved the tasks by hearing as a main source of information and by communicating; they also used the interior dialogue (what one says to oneself) with the very important role in consolidating the system of values; they were very sociable, centered on communication, and they did not orientate so well in space, but they were very precise in actions that did not require too many details. These students were the ones who organized evenings of radio and audiobooks.

The ones who preferred module 4, degree score of 1.55 (centered on the dominant digital sensory canal), stood out by planning tasks and analyzing each detail through their thought process without needing visual or auditory images. They perceived the information that they understood with the help of logic, being very focused and detached from the environment.

*3.2. Stage II Factor Analysis*

In stage II we designed a model based on Confirm Factor Analysis. The model restrictions were determined by 1 formative construct (PES = physical education and sports) and 2 reflective constructs (sensory activity; environment and sports). Our analysis measures the impact of each subitem/factor (loading factor = LF) but does not help us to establish the direction of the influence [38,39].

The SmartPLS software will estimate the model saturation based on a series of indices that enhance how well the model explains the variables and fits the hypothesis established. The relevance of the latent constructs designed was analyzed with Cronbach's alpha test, and the consistency of the model was evaluated with composite reliability, rho A, and average variance extracted (AVE). [40,41].

Our analysis is based on 3 variables (Figure 3 and Table 2):

- PES—a formative variable with 8 items, emphasizing the benefits brought by PES;
- Env Sport—a reflective variable with 2 items, emphasizing the type of sport and the environment of practice (urban/rural);
- Sensory activity—a reflective variable defined by the 4 types of activity (visual, auditory, kinesthesis, and digital) preferred by students.

**Table 2.** Variable analyzed.

| Var Label | Var Subitems | Variable Definition | LF |
|---|---|---|---|
| Sensory activity | Visual | Activity 1 centered on the dominant visual sensory canal | 0.975 |
| | Auditory | Activity 2 centered on the dominant auditory sensory canal | 1.086 |
| | Kinesthetic | Activity 3 centered on the dominant kinesthetic sensory canal | 0.941 |
| | Digital | Activity 4 centered on the dominant digital sensory canal | 0.734 |
| Environment and Sports | Env | Environment: 1-urban or 2-rural | 0.406 |
| | Sport | Type of sports practiced by children | 0.29 |
| PES Role | Health | PES role: sanitation | 0.932 |
| | Fitness | PES role: fitness | 0.774 |
| | Body | PES role: body shaping | −0.17 |
| | Learn | PES role: assimilation of movement concepts | −0.811 |
| | Develop | PES role: psychosocial development | −0.882 |
| | Relax | PES role: recreational activities | 0.287 |
| | Tolerance | PES role: promotes tolerance | 0.266 |
| | Discrim | PES role: cultural discrimination | 0.42 |

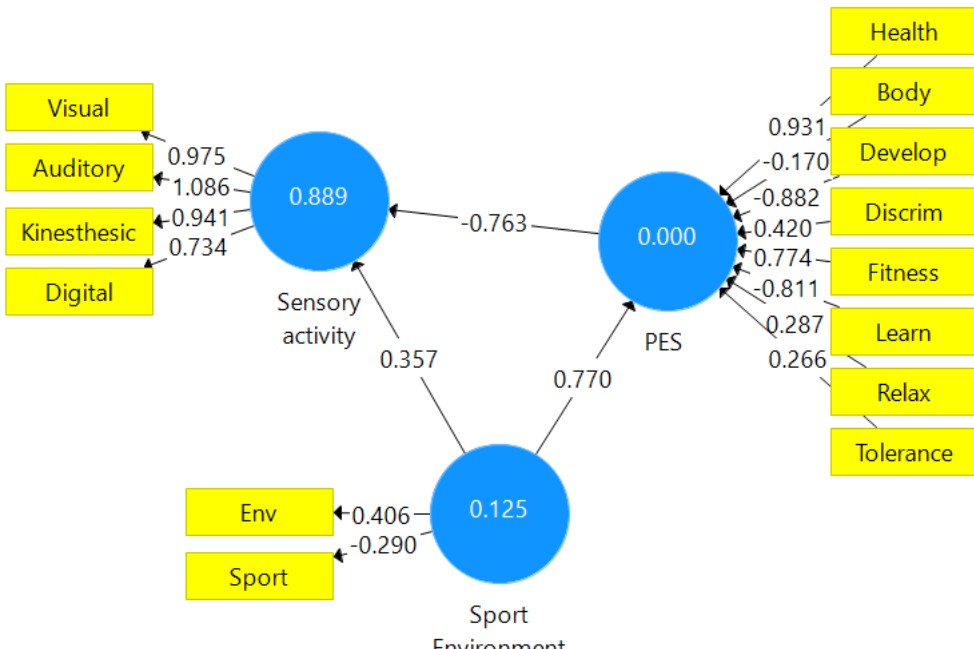

**Figure 3.** Cronbach's alpha coefficients and path analysis.

### 3.2.1. The Stage II Hypotheses of the Research Are

**Hypothesis 2 (H2).** *The students' preferences for the outdoor types of activity (Visual, Auditory, Kinesthesis, and Digital) do not depend on the students' environment provenience (rural or urban), but on the benefits that particular activity brings.*

**Hypothesis 3 (H3).** *The preference for different sports or outdoor activities has a strong and positive impact on PES outputs.*

As a result of the aforementioned questionnaire, based on the effectuated analysis, one does not see the environment the students are coming from, whether urban or rural, as a definitory factor in choosing a certain sport or a certain outdoor activity. The 4 sensory canals are a means for the students to make their choice regarding a certain outdoor activity based on their liking, their enjoyment of that particular action, and the actual perceptible feelings related to sensation [22].

Furthermore, the benefits taken out of these activities, such as communicating better, experimenting with different situations that require planning or organizational skills, experiencing different emotions through movement, etc., all have triggered needs for growth and personal development outside the routine (inside activities), actually in a new routine—that of nature [22].

It is already common knowledge that engaging in activities in the natural environment brings about quite a lot of health benefits, one of the first and most important ones being precisely the outside factor, the fresh air. Though it might be one of the most essential factors, it is completed by the students' willing to step out of their comfort zone and try different things, as the study results have provided. The students have scored very well in their sensory category, assimilating and implementing their perceptive skills in choosing their preferred outdoor activity.

### 3.2.2. Stage II Results

**Construct Reliability and Validity**—SmartPls software provides many tests that can be used to ensure a coherent analysis and interpretation of data and to assume the research outputs. For example, the consistency of our model was grounded on the validation steps provided in Table 3 [38,39]. All the considered variables present very high values for

composite reliability: Cronbach's alpha and rho_A (>0.7—the bottom value authorized), and average variance extracted (AVE) (>0.5—the bottom value approved), meaning that convergent validity can be assumed. These results empower us to believe that all our hypotheses are validated to different extents (Table 3, Figure 3).

**Table 3.** Validation steps/tests.

| Variable | Cronbach's Alpha | Rho_A | Composite Reliability | Average Variance Extracted |
|---|---|---|---|---|
| | >0.7 | >0.7 | >0.7 | >0.5 |
| PES | | 1 | | |
| Sport and Environment | 0.267 | 1 | | |
| Sensory Activity | 0.968 | 0.984 | 0.969 | 0.889 |

The loading factors (LFs) for Sensory Activity latent constructs in Table 2 and Figure 3 enhance that all the items that form sensory activity are very well represented in the model. The visual, auditory, kinesthesis, and digital activities all have loading factors greater than 0.6, meaning that these 4 items evaluate very well student preferences for a different kind of activity.

The loading factors (LFs) for Sport and Environment latent constructs in Table 2 and Figure 3 are less than 0.6. The ENV (LF = 0.406) is important and has a positive influence on the PES role. Outdoor sports practiced in the natural environment have a greater impact on human health, fitness, and relaxation, than inside sports [39,40].

This could very well be a hypothesis proven by simple logic but based on the effectuated research and the scores the sample of 100 students have provided based on their sport or activity of choice, we can surely state that, while inside sports do also have their share of positive aspects, the outdoor activities engulf, through the 4 sensory canals experience, the best effects on human health, physical and psychological development and relaxation.

The type of sport has also a positive influence on the PES roles, but a smaller one (LF = 0.290). For a better model, the Sport and Environment latent constructs should contain some other subitems, such as: arrangements dedicated to each type of sport, integration of discoveries in the field of sports in its practice, and integration of technology facilities in sports training. These are students' suggestions to a question with an open answer. This is the reason that Cronbach's alpha has a small value (0.267) for this construct.

PES role is defined especially by Health (LF = 0.931), Fitness (LF = 0.774), Discrimination (LF = 0.420), and Tolerance (LF = 0.266). Students do not consider PES important for body shaping, personal development, and learning. We have to emphasize that this is a subjective opinion.

**Discriminant Validity**—Our model is statistically robust, as the Fornell–Larcker criterion and Heterotrait–Monotrait criteria are met because all values obtained are equal or less than 0.70 (Table 4) [39,40]:

**Table 4.** Discriminant validity.

| Variable | Fornell–Larcker Criterion | | | Heterotrait–Monotrait Criterion |
|---|---|---|---|---|
| | PES | Sport Environment | Sensory Activity | Sensory Activity |
| PES | | | | |
| Sport and Environment | 0.700 | 0.353 | | 0.398 |
| Sensory Activity | −0.035 | 0.943 | 0.700 | |

**The Path analysis** presents the value below:

- Sports Environment → Sensory activity (0.945). The environment is important for students and has a positive influence on the PES role. Depending on the environment in which a sport is practiced, other types of senses are involved and students' preferences change.

- Sports Environment → PES (0.770). The outdoor sports, practiced in the natural environment have a greater impact on human health, fitness, and relaxation/recreation, than inside sports.
- PES → Sensory activity (−0.763). With a negative value, we may affirm that the role of PES does not influence the students' preference for a different kind of sensory activity. In other words, students consider PES important, no matter what the type of activity.

In Table 5 we may observe a strong positive correlation between the PES role and the environment, empowering H3 of our research. Outdoor sports are preferred by the students because of their positive influence on their physical and mental health. Another positive small correlation can be observed between the Sport and Environment and Sensory Activity. Outdoor sports need different sensory activities. Between PES and Sensory Activity, no correlation is presented. For students it is more important to practice sports than to choose a special type of sensory activity.

**Table 5.** Variable correlation.

| | Latent Variable Correlation | | | R Square | R Square Adjusted | F Square | |
|---|---|---|---|---|---|---|---|
| **Variable** | **PES** | **Sport and Environment** | **Sensory Activity** | | | **PES** | **Sensory Activity** |
| Sensory Activity | −0.035 | 0.357 | 1 | 0.355 | 0.589 | | |
| Sport and Environment | 0.770 | 1 | 0.357 | | | 1.456 | 0.572 |
| PES | 1 | 0.770 | −0.035 | 0.593 | 0.589 | 0.373 | |

The steps presented in Tables 2–5 empower us to assume that our hypotheses are confirmed. The hypothesis tests (SRMR, d_ULS) also have higher estimates for the estimated model than for the saturated model. Thus, we may affirm that our model fits and that H1, H2, and H3 are accepted (Table 6). The standardized root means square residual (SRMR) has a value of less than 0.1, explaining a good fit [38–40]. d_ULS represents the squared Euclidean distance. Thus, our hypothesis is confirmed by a consistent model (Table 6).

**Table 6.** Model Fit.

| Test | Fit Summary | |
|---|---|---|
| | **Saturated Model** | **Estimated Model** |
| SRMR | 0.055 | 0.056 |
| d_ULS | 0.315 | 0.317 |

## 4. Discussion

After the survey results, the objectives have been fulfilled, and as such one can conclude that the outdoor activities from the perception systems point of view can be fitted according to the rank ordering method as follows: the activities included in module 3, centered on the dominant kinesthetic sensory canal, are in first place.

One notices that the majority of students favor a sensory canal, especially in stressful situations. It is important to say that they did not use exclusively a certain sensory canal, but they mainly had a preference and they identified it in selecting certain outdoor activities that would correspond to their needs. The effectuated outdoor activities proved the increase of efficacy in the functioning way of the psychological processes that depend on knowing the dominant representation system and on developing the accessibility of the representation system preferred by students. Knowing the dominant sensory system increases the possibility of knowing the students' preferences out of the four analyzed and numbered sensory systems. The practice has proved the necessity of diversifying the educational programs with outdoor education-specific activities according to the perception

mechanisms, resulting in the attraction of a higher number of students and avoiding the monotony among them.

We have also highlighted the formative potential of the outdoor activities bearing in mind the area of experimented methods ranging from using specific outdoor means to organizing activities depending on the main dominant sensory canal. Finalizing this study allows us to draft the following recommendations:

- Introducing in the curricular aria a program that offers outdoor education activities for students [41,42];
- As a particularity, the outdoor means must be selected depending on the dominant sensory canal of the student group and must be effectuated in real conditions benefiting from the best and most interesting "didactical material—the natural ecosystems or the artificial ones with their terrestrial or aquatic habitats;
- Selecting the specific means that must comprise the improvement of motile skills involved in these types of activities;
- Within the communication sessions, scientific seminars and other manifestations of this type, approaching an outdoor activity is thematic to accomplish an education from the natural value point of view at a national level.
- Sports Environment → Sensory activity (0.945) For students, the environment is crucial. the PES role, and has a favorable impact on it. Other senses may be engaged, and students' choices may change, depending on the environment in which a sport is played.
- Sports Environment → PES (0.770) In comparison to indoor sports, outdoor sports that are played in a natural environment have a stronger positive impact on a person's health, fitness, and leisure time.
- PES → Sensory activity (−0.763) With a negative result, we can state with confidence that PES's function has no influence on pupils' preferences for particular sensory activities. In other words, regardless of the activity, students think PES is important.

## 5. Conclusions

At the end of this research, we would like to accentuate the fundamental idea behind the chosen theme, which is that outdoor activities have a great and positive impact on students' physical and psychological development [2]. We have learned that the majority of students involved in the research have used their preferred sensory systems—visual, auditory, kinesthetic, and digital—in accordance with their correspondent needs when choosing a favorite outdoor activity [2]. Based on this, the research has proved that one needs to further diversify educational programs with outdoor education activities, to achieve a higher number of students involved in such activities and to keep away from day-to-day routine and dullness [21,26,27,30].

We also observed that the students' preferences for the outdoor types of activity do not depend on the students' environment provenience (rural or urban), but on the benefits that particular activity brings. It also depends on the student's limits in effectuating it and has a strong and positive impact on PES outputs [3].

The research has organized activities based on the dominant sensory canal by using specific outdoor means of the undertaking, using the scene in real conditions; these conditions were terrestrial or using water. This has led to a list of recommendations we think are suitable for the upcoming generations, such as implementing in the curricula a program that comprises different means of outdoor education, sorting out the best ways that motile skills can be encapsulated in the different types of activities and organizing communication sessions, seminars and other types of theoretical activities based on the notion of outdoor education, which can further extend and persuade the faculty and the students to perceive education from the natural point of view as one of the most rewarding types of education for the future [18,30].

The confirmation of the three hypotheses leads us to state that the application of the online questionnaire in the direction of the specific and relevant indicators of the research

process makes it possible to highlight the benefits provided by different sports and influence the type of activity (visual, auditory, kinesthetic, and digital).

Limitations: In this paper, there are still some limitations:

Our study employed a small number of subjects and expanding research could target many geographical areas and universities' cycles. This was preliminary research that will be further extended to national level in future research. Another objective of our next study is to evaluate what kind of abilities can be enforced and taught through different kinds of activity (visual, auditory, kinesthetic and digital).

**Author Contributions:** Conceptualization, V.M., G.S., I.T., C.C., D.G.P. and F.N.; methodology, V.M., F.N., G.S. and I.T.; software, C.C. and D.G.P.; validation, F.N., C.C., V.M. and F.N.; formal analysis, C.C. and D.G.P.; investigation, F.N., V.M.; resources, G.S. and I.T.; data curation, V.M.; writing—original draft preparation, F.N., G.S. and V.M.; writing—review and editing. G.S., I.T. and F.N.; visualization, C.C., D.G.P., V.M.; supervision, F.N., I.T., V.M., and G.S.; project administration, V.M. All authors have read and agreed to the published version of the manuscript.

**Funding:** This research received no external funding.

**Institutional Review Board Statement:** Ethical review and approval were waived for this study due to the fact the respondents gave their consent in using the research results.

**Informed Consent Statement:** Ethical review and approval were waived for this study, due to the fact that survey was anonymous, and the respondents agreed that researchers use their answers/opinions for analysis.

**Data Availability Statement:** Not applicable.

**Conflicts of Interest:** The authors declare no conflict of interest.

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
