# Peer review of "The Multiplicative Effect Interaction between Outdoor Education Activities Based on the Sensory System"

_sustainability, doi:10.3390/su141911859_

Round 1

Reviewer 1 Report

The research seems up-to-date and interesting in terms of its subject. Although the research title is not too long, it is suitable for the purpose.

It was not good that the abstract of this research was in the form of a structured summary. In other words, it would be good to write a summary in the summary that includes the purpose, method, data collection tool and summary of the results.

The introduction of the research is appropriate in terms of literature. The introduction part of the research is sufficient in terms of subject area. The bibliographies used are up-to-date. Therefore, the use of new bibliography in the introduction and discussion sections of the research has enriched the research.

Purpose and sub-objectives were written in line with the findings.

The research method is well written. The data collection tools used in the research and their validity and reliability should be emphasized. Care was taken to write the tables used in the research in the form of APA6 standard.

Discussion, conclusion and recommendations sections were written in the research. Overall I would say it is a good article.

Reviewer 2 Report

1. The purpose of the research shall be clearly stated in the Abstract.

2. Minor deficiencies in the reference to sources e. g. in line 76: 4 are the following quotations in form [12,13,14,15] and in line 93 as [17-20]. Standardize the description according to the editorial recommendations. Another example is the absence of spaces between Table and 6 on line 466.

3. In my opinion, it is unclear to include further research hypotheses (H1 and separately H2/H3) in the different parts of the work. H1 (lines 233-235) follows the method and is unobjectionable in this order. From line 299, the authors show the results before presenting further hypotheses (lines 378-382). In addition, the results appear twice: first in line 299 and then as Research and next as Research Results in line 401.

4. The title of Table 3 is located below the table, in contrast to the other 5 titles arranged above the tables.

5. The Conclusions should review the previously established research hypotheses in a transparent manner.

6. Minor editorial errors are displayed in the references.

In line 554 between M and R (the first letters of the author’s name) there are spaces as opposed to line 561, where there are spaces.

In line 570, the abbreviation Int. J. of Obes. Meanwhile, the form of the Int. J. Obes.

Line 601 is J. Ev. Ed, I think it should be J. Env. Ed.

In line 632 there are no dots after the letters of the names of the authors Xu and Niu. There’s DB. and XH, and probably it should be D. B. and X. H. or maybe D.B. and X.H.

There are more similar errors in the references.

Reviewer 3 Report

First of all, I would like to congratulate the authors for the effort put into the manuscript. I think it is very interesting and will be a good starting point for other researchers focusing on this field of outdoor activities. The following comments are intended to help increase the quality of the paper.

The sections are well structured although the numbering of the sections needs to be revised. For example, in the Introduction section there are two subsections numbered 1 (lines 33 and 103). It is advisable to use a numbered list to avoid these issues.

English grammar must be reviewed. Also, the literal translation of some concepts. For example, "orienteering" instead of "orientation". Similarly, it would be necessary to differentiate between the two activities called "hiking" (could it be "trekking and hiking" in line 263 and "mountaineering" in line 264?).

The Introduction section (lines 42-44) mentions that there are several main reasons or motivations for doing outdoor activities (besides self-improvement). It would be necessary to mention them, incorporating some references that can confirm these reasons.

Finally, in accordance with the Journal's editorial line, it would be necessary to include bibliographical references in the Conclusions section to support them. Likewise, although it is not a mandatory section, a commentary on the limitations found in the study could be added.

Round 2

Reviewer 2 Report

The authors' explanations are sufficient for me. I accept manuscript in present form.

Reviewer 3 Report

 Received the new version, thanks for the modifications. In my opinion and experience, the concept "Mountaneering" should not be accompanied by the word "hiking". But I understand that you want to differentiate it from "alpinism" or "mountaing climbing". In this case, it could be left as follows.